# Whole Goat Milk-Based Formula versus Whey-Based Cow Milk Formula: What Formula Do Infants Enjoy More?—A Feasibility, Double-Blind, Randomized Controlled Trial

**DOI:** 10.3390/nu15184057

**Published:** 2023-09-19

**Authors:** Camille Jung, Adolfo González Serrano, Christophe Batard, Elisa Seror, Georges Gelwane, Amélie Poidvin, Isabelle Lavallée, Annie Elbez, Maxime Brussieux, Colin Prosser, Sophie Gallier, Marc Bellaïche

**Affiliations:** 1Clinical Research Center, Centre Hospitalier Intercommunal de Créteil, 94000 Créteil, France; 2Inserm, IMRB, Université Paris-Est-Créteil, 94000 Créteil, France; adolfo.gonzalez@chicreteil.fr; 3Private Pediatric Practice, 94080 Vincennes, France; 4Private Pediatric Practice, 75000 Paris, France; 5Private Pediatric Practice, 92012 Boulogne-Billancourt, France; 6Private Pediatric Practice, 92035 La Garenne-Colombes, France; 7Private Pediatric Practice, 94700 Maisons-Alfort, France; 8Dairy Goat Co-Operative (N.Z.) Ltd., Hamilton 3204, New Zealandsophie.gallier@dgc.co.nz (S.G.); 9Department of Pediatric Gastroenterology, Robert Debré Hospital, Assistance Publique-Hôpitaux de Paris, 75019 Paris, France; bellaichemarc@gmail.com

**Keywords:** whole goat milk infant formula, cow milk infant formula, food enjoyment, infant feeding behavior, baby eating behavior questionnaire

## Abstract

(1) Background: While goat milk formula (GMF) is an alternative to cow milk formula (CMF), infants’ preferences for one over the other have not been formally assessed. Specifically, our aim in this study was to determine whether infants experience fewer feeding behavior problems with whole milk-based GMF than with conventional whey-based CMF. (2) Methods: This was a multicenter, double-blind, randomized controlled trial with two-arm parallel assignment conducted in six pediatricians’ offices in or near Paris, France, between June 2018 and 31 December 2021. Overall, 64 healthy infants (≤4 months old), predominantly formula-fed, were randomly assigned to either the whole milk-based GMF (*n* = 33) or whey-based CMF (*n* = 31) arm. Parents completed the Baby Eating Behavior Questionnaire (BEBQ) and the modified QUALIN questionnaire to evaluate infant feeding behavior and quality of life (psychomotor and socioemotional development), respectively, at inclusion (1 to 5 days before milk delivery) and the final visit (day 28 ± 3 after milk delivery). Informed consent was obtained for all recruited patients, and an ethical committee approved the study. (3) Results: Changes in BEBQ Enjoyment of Food and Slowness in Eating subscale scores from inclusion to final visit did not differ between arms. However, there were significant improvements in subscale scores for Food Responsiveness (GMF: 0.15 ± 1; CMF: −0.48 ± 0.81; *p* = 0.010) and General Appetite (GMF: 0.26 ± 1.2; CMF: −0.48 ± 0.88; *p* = 0.012), and modified QUALIN (GMF: 4.6 ± 9.4; CMF: −0.40 ± 7.6; *p* = 0.03) scores in favor of the GMF group. (4) Conclusions: In this double-blind, randomized controlled trial, GMF-fed infants exhibited a greater general appetite than CMF-fed infants, possibly due to differences in the composition of these formulas (i.e., protein and lipid profiles). In addition, GMF-fed infants enjoyed a better quality of life. There was no difference in food enjoyment between groups. These findings suggest that whole-milk-based GMF could be an attractive alternative to whey-based CMF. Clinical trial registration: NCT03488758 (clinicaltrials.gov).

## 1. Introduction

During the first weeks of life, changes in infant formula for bottle-fed infants are frequent [1]. Apart for cow’s milk protein allergy, this may be due to functional gastrointestinal disorders (FGIDs) or a lack of appetite in the infant. Indeed, up to 55% of infants exhibit prolonged crying, fussing, gassiness, abdominal distension, regurgitation, reflux, diarrhea, or constipation in the first 6 months of life [2]. If recurrent, these symptoms can significantly reduce the quality of life for those children affected [3]. Prolonged crying in infants also significantly impacts family dynamics through disruption of sleep and family routines and increased parental anxiety, stress, and frustration [4].

Although breastfeeding is the most appropriate way to feed infants in the first months of life, most infants are no longer breastfed after 6 months [5]. In France, breastfeeding is declining [6], resulting in partial or exclusive infant formula feeding. In cases of FGIDs, it is widely held by parents and physicians alike that changing formula can help relieve gastrointestinal (GI) discomfort in formula-fed infants. FGIDs are often transient and resolve over time but, as they impact the quality of life of infants and families, parents want a quick solution, which often leads to a change in formula for formula-fed infants [1]. Up to 50% of parents report partial or complete resolution of feeding intolerance after formula replacement [7], and ~50% of infants switch to another cow’s milk-based or alternative formula during the first 6 months of life [8], mainly due to digestive discomfort (i.e., regurgitation, colic, or fussiness). Despite the high prevalence of feeding difficulties in infants, there is no objective assessment that clinicians can use in clinical routine to test responses to different formulas and determine whether switching to a new formula impacts feeding outcomes.

Goat milk is an alternative source of protein for infant formula. Hence, goat milk formula (GMF), either based on whole goat milk or whey based, offers an alternative to whey-based cow milk formula (CMF) [9]. Because they differ in composition and physicochemical properties, goat milk and cow milk may not have the same digestive and metabolic effects in children. Infants and young children might better tolerate, and consequently prefer, formula made from one or the other dairy source [10,11].

Taste development in early life is complex and still poorly described [12,13]. The sensory characteristics of infant formula are a key factor driving acceptance by the formula-fed child. However, to date, most studies have investigated the taste acceptance of conventional CMF vs. soy-based formula or hydrolysate formulas [12,14,15]. Except for the demonstration of adequate growth support as an indicator of acceptance and sufficient consumption [16], there is a lack of understanding whether GMF and CMF are enjoyed similarly and if one is better accepted than the other.

Several randomized and non-randomized trials have compared growth—i.e., changes in weight, length, and head circumference—in children fed a CMF or a GMF. A recent double-blind, randomized controlled trial (RCT) by He et al., comparing a whey-based GMF, a conventional CMF, and breastfeeding in the first 4 months of life, found similar levels of growth between the CMF and GMF groups [17]. Indeed, after 16 weeks of the intervention, the average weight for infants fed GMF was 7009.0 g (with a standard error of 96.1), whereas infants fed CMF had an average weight of 6781.2 g (with a standard error of 99.5), and breastfed infants weighed, on average, 6449.3 g (with a standard error of 139.9). When analyzing the data in both intention to treat (ITT) and per protocol (PP), taking the initial weight into account as a covariate, the authors found that the weight gain in the GMF group was not significantly lower than that in the CMF group. In addition, GI disorders (reflux, flatulence, fussiness, or colic) were highly prevalent (≥80% of infants) in both the CMF and GMF groups, and infants fed the whey-based GMF were more likely to pass hard stools than those fed the CMF. A longer-term RCT by Zhou et al., where infants were either breastfed or received a whole milk-based GMF or a conventional CMF for the first 12 months of life, reported similar growth and nutritional outcomes, stool patterns and ease of settling in both formula-fed groups [16]. In another RCT, Grant et al. reported comparable frequencies of FGIDs in infants fed either a whole milk-based GMF or a conventional CMF, though GMF-fed infants had more daily bowel movements [18]. Thus, the authors randomized 72 infants to be fed either GMF or CMF infant formula. The median number of bowel movements per day was higher in the GMF group compared to the CMF group (2.4 [1.1–4.0] vs. 1.7 [1.0–4.4], *p* = 0.01). However, the number of constipated patients did not differ in the two groups (4 (12%) vs. 2 (6%), *p* = 0.35, respectively). Similarly, the number of infants with prolonged crying was similar in both groups (3 (9%) in the GMF group vs. 7 (19%) in the CMF group, *p* = 0.19). This was similar to results from an observational study in the first 12 months of life where infants who receive partially or exclusively a whole milk-based GMF had stool patterns closer to those of breastfed infants than infants who were fed a conventional CMF [19]. Xu et al. did not report any differences in stool patterns or duration of crying in infants fed for 4 months either a whey-based GMF or CMF [16,20]. Moreover, a recent meta-analysis including 4 RCTs concluded that GMFs, compared to CMFs, exhibited a favorable safety profile and were well tolerated [21]. Thus, while GI discomfort in infants on whole milk-based or whey-based GMF versus CMF has been considered to an extent, as a secondary endpoint in several RCTs, infant preferences for one type of formula over the other have not been compared.

Accordingly, in the present study, we compared the feeding behavior and quality of life of infants fed either whole milk-based GMF or whey-based CMF and measured degrees of GI discomfort for these study groups.

## 2. Materials and Methods

### 2.1. Design and Participants

This trial was a double-blind RCT with two-arm parallel assignment conducted at six pediatrician’s offices in the greater Paris area of France. An independent ethics committee approved the study protocol (CPP Sud Méditerranée 3, France; ref.: 2018.01.01), which complied with the Declaration of Helsinki. Both legal representatives of each patient provided their written informed consent before inclusion. The study was registered prospectively at ClinicalTrials.gov (NCT03488758).

Eligible patients were healthy infants ≤4 months old who were predominantly formula fed and followed by a private-practice pediatrician. Patients were ineligible if born at a gestational age of <37 weeks, had a confirmed or suspected cow’s milk protein allergy, were predominantly breastfed (≥50% of feeding volume), or suffered from chronic diseases.

At trial initiation, participants were randomly assigned to the GMF or CMF groups at a 1:1 ratio, applying a double-blind procedure. After checking inclusion/exclusion criteria, the investigators explained the study to the legal representatives at the screening visit. The clinical and demographic characteristics, and relevant medical and maternal history, of participating individuals were recorded at the inclusion visit (baseline), which took place 1 to 5 days before milk delivery. Outcome ascertainment and collection of safety assessment data were performed at trial centers during the inclusion visit and the final visit (day 28 ± 3 after milk delivery). Participants’ parents were contacted via phone on day 14 ± 3 to verify compliance and safety assessments. Compliance at days 14 and 28 was defined as having maintained exclusive bottle feeding (with or without additional breastfeeding) with the study formula delivered.

### 2.2. Study Products

The whole milk-based GMF (20:80 whey:casein ratio and about 48% milk fat; Capricare with added DHA and arachidonic acid) or whey-adjusted CMF (60:40 whey:casein ratio and about 22% milk fat) were produced by Dairy Goat Co-operative (N.Z.) Ltd., Hamilton, New Zealand. The composition (see Appendix A) of both infant formulas was identical and complied with the applicable European regulation [22]. The formula was delivered to the families within the 48 h following randomization. The cans of formulas included the same instructions for preparation and feeding guide. Participants continued to be fed formula until day 28 ± 3 (end of study). When breaches of the study protocol were identified, the concerned participants were still followed-up for outcome ascertainment.

### 2.3. Outcome Measures

The primary outcome measure was the change in food enjoyment, assessed as the difference in BEBQ Enjoyment of Food subscale scores between the inclusion visit (baseline) and final visit (day 28 ± 3). Secondary outcome measures were the other BEBQ subscale scores; Montreal Children’s Hospital Feeding Scale (MCH-FS) scores; quality of life, as reflected by modified QUALIN scores; GI symptoms; and anthropometric variables.

The BEBQ, developed in 2011 by Llewelyn et al. [23], is an 18-item parent-report psychometric instrument measuring appetite in infants while exclusively fed milk. It was derived from the Children’s Eating Behavior Questionnaire validated for older ages. The BEBQ includes the following subscales: Enjoyment of Food, Food Responsiveness, Satiety Responsiveness, and Slowness in Eating, as well as an independent item reflecting General Appetite. The BEBQ was validated in a large cohort of twins, including 2402 families. This validation confirmed good internal reliability for each subscale, with Cronbach’s alpha values of 0.81, 0.79, 0.76, and 0.73, respectively, for the subscales ‘enjoyment of food’, ‘food responsiveness’, ‘slowness in eating’, and ‘satiety responsiveness’. Additionally, the correlation between the item “general appetite” and the four subscales was also strong.

Designed by pediatricians and healthcare professionals, the MCH-FS is a 14-item parent reporting instrument for identifying feeding difficulties in children [24]. Scores range from 35 to 102 points and are interpreted as follows: <61, no feeding difficulty; or 61–65, mild; 66–70, moderate; and >70, severe feeding difficulty.

At baseline and the final visit, health-related quality of life was assessed using a modified version of the 34-item QUALIN questionnaire (French version), intended for children 3 months to 3 years old and completed by parents or caregivers [25]. This modified version (Appendix A) omitted 14 of the 34 items in the original French questionnaire that were not suited to infants <3 months old, and only the overall QUALIN score was calculated. In addition, it was asked the parents to score the formula liking using a four-level visual analog scale in the final visit.

Data on adverse events during the study period were recorded during each participant visit. Safety was evaluated in all participants who underwent randomization and received ≥1 serving of formula.

### 2.4. Statistical Analysis

The number of infants needed for detecting a statistical difference was calculated based on differences in the BEBQ Enjoyment of Food subscale reported by Llewellyn et al. [23]. Thus, it was determined that to detect a statistical difference, 78 infants were needed for each group—assuming a difference between groups of 0.27 in BEBQ Enjoyment of Food subscale scores with a standard deviation (SD) of 0.6, a power of 80%, and an alpha level of 0.05. However, because this was a feasibility study with limited data allowing for a precise calculation of the number of subjects to include, and due to the difficulty of including such a large number of healthy infants, planned enrollment for each group was lowered to 50, reducing the power to 60%.

Summary statistics described the demographic and clinical characteristics of participants. Categorical data were described using frequencies and percentages. Quantitative data were described by the mean and standard deviation (SD).

The ITT population was included in the primary outcome analysis. Participants from parents who refused to participate just after randomization and refused the milk delivery were excluded from the study. Changes in BEBQ and modified QUALIN scores for the two groups were compared using Student’s *t*-tests or Wilcoxon tests, depending on score distribution. Differences between groups were calculated and reported with their SD. Categorical data were compared using the chi-square or Fisher’s exact test, as appropriate.

Sensitivity analyses were performed for the PP population, composed of all participants who underwent randomization but for whom no severe non-compliance to the intervention was observed. Analysis of outcomes for the PP population applied the same methods as for the ITT population.

All statistical tests were two-sided. The threshold for statistical significance was set to *p* < 0.05. SAS 9.4 for Windows (SAS Institute, Cary, NC, USA) was used for statistical analysis. Incomplete data on event occurrence dates were imputed as means.

## 3. Results

Between June 2018 and December 2021, 70 patients were screened for eligibility. Due to the COVID-19 pandemic, inclusions were suspended between March and June 2020, as non-urgent medical appointments were impossible then. Then, due to the expiry date of the study milk cans, inclusions were stopped in December 2021. A total of 5 of the 70 screened patients declined to participate, and family consent was withdrawn for one other just after randomization. The ITT population consisted of 64 participants: 33 in the GMF group and 31 in the CMF group. Of these, 3 in the CMF group and 1 in the GMF group stopped treatment due to regurgitation or fussing (Figure 1). The PP population consisted of 60 participants: 32 GMF and 28 CMF infants.

Demographic and clinical characteristics were similar between groups (Table 1).

Differences in five BEBQ subscale scores between baseline and day 28 ± 3 for the CMF and GMF groups were compared (Table 2). No differences were detected for the Enjoyment of Food and Slowness in Eating subscales. However, in contrast to CMF children, the GMF group had improved Food Responsiveness and General Appetite scores. Indeed, while Food Responsiveness increased in the GMF group, it decreased in the CMF group, in both the ITT (GMF: [mean ± SD] 0.15 ± 1.0; CMF: −0.48 ± 0.81; *p* = 0.010) and PP (GMF: 0.14 ± 1.05; CMF: −0.42 ± 0.76; *p* = 0.04) analyses. Similar results were found for the BEBQ General Appetite item, again in both the ITT (GMF: 0.26 ± 1.2; CMF: −0.48 ± 0.88; *p* = 0.012) and PP (GMF: 0.31 ± 1.3; CMF: −0.5 ± 0.95; *p* = 0.012) analyses. On the other hand, Satiety Responsiveness improved in the CMF group but not the GMF group, according to the ITT analysis (GMF: −0.15 ± 0.93; CMF: 0.31 ± 0.88; *p* = 0.049). However, no difference for this subscale was detected in the PP population.

QUALIN scores (Table 3) at the end of the study (i.e., after 28 days ± 3 of formula feeding) were significantly higher in the GMF group than the CMF group in the ITT (GMF: 34.5 ± 6.7; CMF: 30.0 ± 7.3; *p* = 0.02) and the PP (GMF: 34.4 ± 6.7; CMF: 30.2 ± 7.4; *p* = 0.02) populations. QUALIN scores improved for the GMF group from baseline to the end of the study but not for the CMF group. At final visit, in the ITT population, there were no observed differences between groups in terms of skin lesions or GI symptoms (Table 4), except for parental reports of constipation, which was more frequent among GMF infants (GMF: 38.7%; CMF: 14.8%; *p* = 0.042).

According to MCH-FS scores, 2 (6%) GMF children had mild feeding difficulties and 1 (3%) CMF child had a moderate feeding difficulty at the end of follow-up (*p* = 0.22).

Finally, at the end of the study, parents were asked to use a four-level visual analog scale to evaluate how much their children liked the formula they were fed, and ratings did not differ between formula types (Figure 2).

Overall, 10 non-serious adverse events were reported during the study period. In the CMF group, one patient had a fracture after a trauma, and another experienced bronchiolitis. In the GMF group, 3 patients suffered from constipation, 3 had viral gastroenteritis and 1 patient a urinary infection. None of these adverse events were linked to the study products by the investigators.

## 4. Discussion

This double-blind RCT is, to our knowledge, the only one to compare feeding behaviors and preferences of infants fed either with GMF or conventional CMF. We observed similar levels of food enjoyment, i.e., subjective degree of pleasure experienced during feeding, for both types of formula. However, infants fed GMF, unlike those fed CMF, showed improved food responsiveness, general appetite, and quality of life.

These findings are somewhat similar to those for animals and human adults. Klockars et al. [26] showed that cow and goat milk are very palatable to food-deprived or sated rats and mice. However, when these animals chose between the two kinds of milk, they preferred goat milk. The study’s authors demonstrated that this preference was associated with increased gene expression in reward mechanisms. In humans, a study of 33 adults noted greater satiety after a breakfast featuring caprine dairy products than one with bovine dairy products [27]. Furthermore, participants’ blood concentrations of appetite-related hormones, including ghrelin and GLP-1, measured at various time points, suggested that GLP-1 may be involved in the satiating effect of caprine dairy products.

Infant formulas, e.g., CMF or GMF, must adhere to strict regulations governing their protein, carbohydrate, lipid, mineral, and vitamin content [28]. They cannot replace human breast milk, which best meets the nutritional requirements of newborns and infants in their first months of life. However, when breastfeeding is impossible, marketed infant formula must also meet these requirements to ensure optimal growth and development. Like previous RCTs that compared the growth and nutritional status of infants fed with GMF or CMF [17,20], our study noted similar weights and lengths for both groups at day 28. In the study by Zhou et al. [16], compliance (measured as exclusive milk feeding to 4 months of age) in the GMF group was similar to that in the breastfed group and higher to that in the CMF group. This higher compliance may have been related to higher avidity to consume the GMF.

Beyond their general composition, dairy-based infant formulas vary by source of milk ingredients (bovine or caprine and whole milk or skim milk or whey protein concentrate) and types of lipids added (vegetable oils alone or mixed with milk fat) [29]. Our RCT compared a GMF made with whole goat milk (supplying 50% of formula lipids as milk fat) and vegetable oils to a whey-based CMF made with whole cow milk and whey protein concentrate and vegetable oils. Several studies considering the composition of infant formulas made with whole goat milk suggest they have physicochemical properties that are better for infants than conventional CMFs [30]. In particular, in vitro studies have observed faster gastric digestion of whole goat milk infant formulas compared to conventional CMFs, likely due to the formation of smaller flocs of aggregated proteins and oil droplets [31]. One study in rodents also revealed quicker gastric emptying for animals whose diet was supplemented with goat milk versus cow milk [32]. Thus, beyond a potential impact on sensorial characteristics of the formulas, the increased proportion of milk in the whole goat milk formula supplies higher levels of sn-2 palmitic acid, Milk Fat Globule Membrane, cholesterol, and short and medium-chain fatty acids, which may influence the digestion of lipids [29,33]. Moreover, preliminary data suggest that whole milk-based GMFs alter gut flora differently from CMFs [34]. These functional properties of formulas made from whole goat milk may partly explain why infants prefer them. In addition, extrapolation of the results to any GMFs should be made with caution, as (i) physicochemical properties during digestion are affected by the level of αs1-casein in the milk, which depends on the genetics of the goats [22], and (ii) some GMFs are whey based and therefore have different protein and lipid profiles and organoleptic and digestive characteristics.

Our study did not detect any difference in tolerance or GI discomfort between the two formulas tested. However, the presence or absence of GI disorders was not a criterion for inclusion, i.e., there was no record at baseline. At the end of the 28 days feeding period, 27.6% of children in the ITT population (for whom data were available) exhibited constipation, with greater prevalence in the GMF group. Nevertheless, most published studies have reported that bowel movement frequency in infants given GMF is similar to or higher than with a CMF diet [16,20,35]. In addition, Infante et al. [35] reported improvement of constipation symptoms in CMF-fed infants after switching to a whole milk-based GMF for 3 weeks. Therefore, the difference in constipation prevalence, based on parental report only, should be interpreted with caution in this study. There was also no difference between groups for cutaneous signs (atopic dermatitis, rash, and dry skin), though very few children developed such lesions.

Our study had limitations. For instance, we lacked the information to precisely calculate the number of subjects required for our study. Thus, we initially planned to include 100 infants. However, we fell short of this number, primarily due to the COVID-19 pandemic, which slowed recruitment and caused difficulty in obtaining parental consent for their child’s participation. On the other hand, those successfully included children were seldom lost to follow-up. Finally, while the infants’ feeding behavior was assessed using a validated scale adapted to their age group, their quality of life was measured with a modified version of the QUALIN instrument, omitting items not suitable for infants <3 months old. Unlike the original version, this modified version has not been validated. In addition, the CMF used in this study was made with whole milk instead of the more traditionally used skim milk and therefore contained some milk fat, which may have different sensory characteristics to a conventional CMF with vegetable oils only as the source of lipids.

Our study also had strengths despite the intricacy of this kind of trial particularly, as parents could be perturbed by not knowing what formula their child was receiving. This situation could limit inclusions. However, we were able to include an acceptable number of patients and the compliance was good. Also, the BEBQ proved to be a useful tool for healthcare professionals to easily determine infant feeding behavior when switching to a new formula. Finally, our study provides valuable estimates for planning further trials assessing food enjoyment in infants.

Although there were no differences in the enjoyment of the different formulas, infants fed with whole milk-based GMF experienced better food responsiveness and quality of life than those fed with whey-based CMF. Also, besides constipation that was more common in GMF-fed infants by parental report, no adverse effect was observed with higher frequency in the GMF group. However, studies with a larger number of subjects and with a longer follow-up duration are needed to confirm these results: trials assessing food enjoyment in infants are feasible, and further similar trials could help to draw firm conclusions about the better enjoyment of whole milk-based GMF over conventional CMF.

## Figures and Tables

**Figure 1 nutrients-15-04057-f001:**
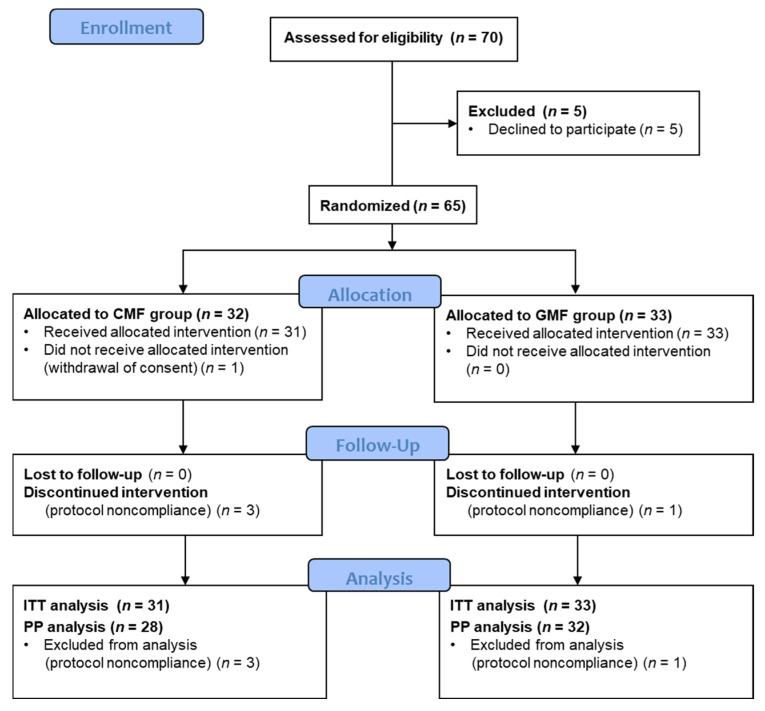
Study CONSORT Flowchart. CMF: cow milk formula; GMF: goat milk formula; ITT: intention-to-treat; PP: per protocol.

**Figure 2 nutrients-15-04057-f002:**
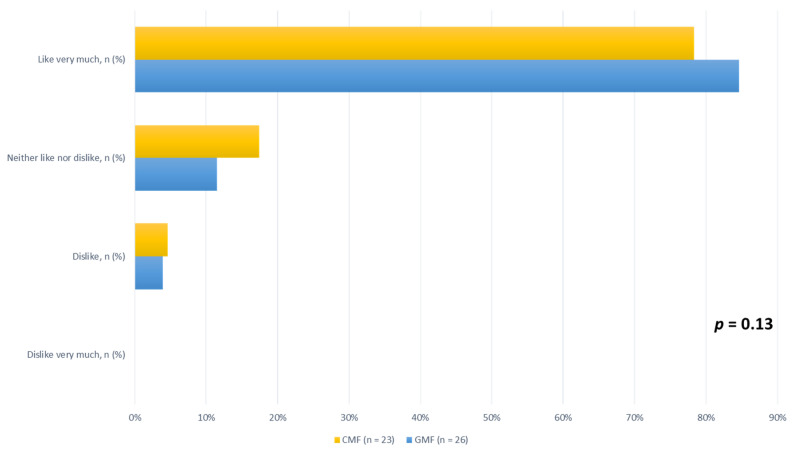
Children’s enjoyment of CMF and GMF, according to four-level visual analog scale ratings, for ITT population at end of study. CMF: cow milk formula; GMF: goat milk formula.

**Table 1 nutrients-15-04057-t001:** Patient demographic and clinical characteristics. CMF: cow milk formula; GMF: goat milk formula; SD: standard deviation; ^^^: Fisher’s exact test; *: Student’s *t*-test; ^µ^: Wilcoxon test. Each value expressed as mean ± SD and as median (interquartile range), number of participants (%), or probability.

Characteristics	Total (*n* = 64)	GMF (*n* = 33)	CMF (*n* = 31)	*p*
Age, months	2.4 ± 1.2 2(1–3)	2.4 ± 1.2 2.5(1.5–3.5)	2.4 ± 1.1 2(1–3.5)	0.97 *
Males	35 (54.7)	20 (62.5)	15 (53.5)	0.33 ^^^
Birth Weight, g	3337.7 ± 700.9 3270(2997.5–3765)	3415.8 ± 861.4 3507.5(3072.5–3877.5)	3248.5 ± 454.0 3165(2937.5–3415)	0.36 *
Length, cm	50.2 ± 2.3 50(48.5–51)	50.4 ± 2.6 50.3(49–51.5)	49.9 ± 2.0 50(48–51)	0.37 *
Inclusion Weight, g	5329.2 ± 1366 5335(4595–6225)	5540.1 ± 1213.2 5625(4650–6195)	5104.6 ± 1500.7 4995(4100–6515)	0.20 *
Length, cm	58.8 ± 4.4 58.8(56–61.8)	58.9 ± 4.4 58.8(56.3–61)	58.7 ± 4.5 58.3(55.5–63.5)	0.85 *
D28 ± 3 Weight, g	6321.6 ± 1105.2 6150(5470–7130)	6314.0 ± 1072.0 6170(5600–6790)	6330.6 ± 1164.8 6095(5400–7200)	0.90 ^µ^
Length, cm	62.0 ± 4.0 62(60–65)	61.8 ± 4.1 62(59–64)	62.3 ± 4.0 63.3(60–65)	0.75 ^µ^
Maternal age, years	34.3 ± 4.8 35(32–36)	34.2 ± 4.1 35(32–36)	35.1 ± 4.9 35(32–39.5)	0.44 *
Smoking during pregnancy Yes No	10 (16.6) 50 (83.3)	5 (15.6) 27 (84.3)	5 (17.8) 23 (82.1)	0.54 ^^^
Alcohol consumption during pregnancy Yes No	1 (1.7) 63 (98.4)	1 (3.1) 31 (96.8)	0 (0.0) 28 (100.0)	1.0 ^^^
Gestational diabetes Yes No	7 (10.9) 57 (89.0)	4 (12.5) 28 (87.5)	3 (10.7) 25 (89.2)	0.72 ^^^
Delivery type Vaginal birth Planned cesarean Unplanned cesarean	51 (79.7) 4 (6.7) 9 (14.0)	25 (78.1) 3 (9.3) 4 (12.5)	22 (78.5) 1 (3.5) 5 (17.8)	0.64 ^^^

**Table 2 nutrients-15-04057-t002:** Changes in BEBQ subscale scores relative to baseline, according to PP and ITT analyses. CMF: cow milk formula; GMF: goat milk formula; ITT: intention-to-treat; PP: per protocol; SD: standard deviation; D0: Baseline (inclusion visit); D28: D28 ± 3 after milk delivery (end of the study); Δ: change; *: Student’s *t*-test; ^µ^: Wilcoxon test. Each value expressed as mean ± SD or probability and as median (interquartile range).

BEBQ Subscales		PP Population				ITT Population			
		Total (*n* = 60)	GMF (*n* = 32)	CMF (*n* = 28)	*p*	Total (*n* = 64)	GMF (*n* = 33)	CMF (*n* = 31)	*p*
Enjoyment of Food	D0	4.4 ± 0.71 4.75(4–5)	4.35 ± 0.85 4.63(4.25–5)	4.47 ± 0.52 4.75(4–5)	0.99 ^µ^	4.40 ± 0.71 4.75(4–5)	4.35 ± 0.83 4.5(4.25–5)	4.45 ± 0.57 4.75(4–5)	0.57 ^µ^
	D28	4.58 ± 0.53 4.75(4.25–5)	4.66 ± 0.47 4.88(4.5–5)	4.50 ± 0.59 4.75(4–5)	0.35 ^µ^	4.55 ± 0.56 4.75(4.25–5)	4.65 ± 0.45 4.75(4.5–5)	4.44 ± 0.65 4.55(4–5)	0.14 ^µ^
	ΔD0 − D28	0.15 ± 0.87 0(−0.25–0.38)	0.30 ± 1.03 0(0–0.75)	−0.01 ± 0.61 0(−0.25–0.25)	0.32 ^µ^	0.15 ± 0.82 0(−0.25–0.3)	0.30 ± 0.98 0(0–0.5)	−0.01 ± 0.59 0(−0.25–0.25)	0.18 ^µ^
Food Responsiveness	D0	2.05 ± 0.76 1.8(1.4–2.3)	1.91 ± 0.67 1.78(1.4–2.2)	2.19 ± 0.83 2(1.4–2.6)	0.13 ^µ^	2.05 ± 0.76 1.9(1.4–2.4)	1.91 ± 0.67 1.75(1.4–2.2)	2.19 ± 0.83 2(1.4–2.6)	0.13 *
	D28	1.89 ± 0.91 1.6(1.2–2.4)	2.07 ± 0.94 1.9(1.4–2.6)	1.68 ± 0.85 1.4(1–2)	0.04 ^µ^	1.90 ± 0.85 1.6(1.2–2.2)	2.06 ± 0.89 1.89(1.4–2.4)	1.71 ± 0.78 1.4(1–1.89)	0.11 *
	ΔD0 − D28	−0.12 ± 0.96 −0.4(−0.6–0.33)	0.14 ± 1.05 0.1(−0.6–0.6)	−0.42 ± 0.76 −0.4(−0.8–0)	0.04 ^µ^	−0.16 ± 0.96 −0.4(−0.6–0.34)	0.15 ± 1.00 0.2(−0.6–0.49)	−0.48 ± 0.81 −0.4(−1–0)	0.010 ^µ^
Slowness in Eating	D0	2.53 ± 1.0 2.25(1.88–3.25)	2.69 ± 0.98 2.5(2–3.38)	2.36 ± 0.98 2.25(1.63–3)	0.22 *	2.56 ± 0.99 2.25(1.88–3.25)	2.72 ± 0.99 2.5(2–3.5)	2.40 ± 0.98 2.25(1.75–3.25)	0.18 ^*^
	D28	2.13 ± 0.92 2(1.25–2.75)	2.11 ± 0.89 2(1.25–2.75)	2.16 ± 0.96 1.88(1.25–3)	0.82 *	2.16 ± 0.88 2.16(1.5–2.75)	2.11 ± 0.84 2(1.5–2.50)	2.21 ± 0.93 2.16(1.5–3)	0.64 ^*^
	ΔD0 − D28	−0.41 ± 0.97 −0.25(−0.75–0.25)	−0.61 ± 1.06 −0.25(−1.25–0)	−0.17 ± 0.81 0(−0.5–0.25)	0.11 ^µ^	−0.40 ± 0.95 −0.25(−0.75–0.25)	−0.59 ± 1.0 −0.25(−1.25–0)	−0.17 ± 0.77 0(−0.5–0.25)	0.08 ^µ^
Satiety Responsiveness	D0	2.63 ± 0.78 2.67(2–3.33)	2.76 ± 0.78 3(2.33–3.33)	2.49 ± 0.77 2.67(2–3)	0.17 *	2.64 ± 0.80 2.67(2–3.33)	2.74 ± 0.78 3(2.33–3.33)	2.53 ± 0.81 2.67(2–3)	0.29 *
	D28	2.70 ± 0.89 2.67(2–3.33)	2.58 ± 0.76 2.67(2–3)	2.83 ± 1.01 2.67(2–3.67)	0.28 *	2.71 ± 0.83 2.69(2.17–3.33)	2.59 ± 0.73 2.67(2–3)	2.83 ± 0.93 2.71(2.33–3.67)	0.24 *
	ΔD0 − D28	0.10 ± 0.95 0(−0.34–0.84)	−0.12 ± 0.94 0(−0.67–0.34)	0.35 ± 0.91 0.33(0–1)	0.06 ^µ^	0.07 ± 0.93 0(−0.5–0.71)	−0.15 ± 0.93 0(−0.67–0.34)	0.31 ± 0.88 0.33(0–1)	0.049 *
General Appetite	D0	3.85 ± 0.96 4(3–5)	3.68 ± 1.05 4(3–4)	4.0 ± 0.8 4(3.5–5)	0.20 ^µ^	3.84 ± 0.97 4(3–5)	3.69 ± 1.03 4(3–4)	4.0 ± 0.90 4(3–5)	0.21 ^µ^
	D28	3.79 ± 1.09 4(3–5)	4.0 ± 1.08 4(4–5)	3.54 ± 1.07 3.5(3–4)	0.08 ^µ^	3.76 ± 1.04 4(3–5)	3.98 ± 1.03 4(3.8–5)	3.52 ± 1.02 3.8(3–4)	0.07 ^µ^
	ΔD0 − D28	−0.07 ± 1.2 0(−1–0)	0.31 ± 1.3 0.00(0–1)	−0.5 ± 0.95 0(−1–0)	0.012 ^µ^	−0.10 ± 1.13 0(−1–0)	0.26 ± 1.23 0(−0.1–1)	−0.48 ± 0.88 0(−1–0)	0.012 ^µ^

**Table 3 nutrients-15-04057-t003:** Change in quality of life from baseline, as measured by modified QUALIN score, for PP and ITT populations. CMF: cow milk formula; GMF: goat milk formula; ITT: intention-to-treat; PP: per protocol; D0: Baseline (inclusion visit); D28: D28 ± 3 after milk delivery (end of study); * Student’s *t*-test. Each value expressed as mean ± SD or probability and as median (interquartile range).

		PP Population				ITT Population			
		Total (*n* = 60)	GMF (*n* = 32)	CMF (*n* = 28)	*p*	Total (*n* = 64)	GMF (*n* = 33)	CMF (*n* = 31)	*p*
Modified QUALIN score	D0	30.4 ± 8.4 32(25–36)	30.4 ± 8.5 31.5(23.5–36.5)	30.4 ± 8.4 32(26–34.5)	0.99 *	30.1 ± 9.2 32(25–36)	29.4 ± 10.1 31(23–36)	30.8 ± 8.3 32(26–35)	0.68 *
	D28	32.5 ± 7.3 33.5(26.5–38)	34.4 ± 6.7 35.5(30–39)	30.2 ± 7.4 31(24–36)	0.02 *	32.4 ± 7.3 33(26–38)	34.5 ± 6.7 35.5(30–39)	30.1 ± 7.3 31(24–36)	0.02 *
	ΔD0 − D28	2.8 ± 8.9 1.5(−3.5–7)	4.6 ± 9.4 3(−1–9)	−0.4 ± 7.6 −2.5(−6–4)	0.03 *	2.2 ± 8.9 1(−4–7)	4.8 ± 9.4 3(−1–9)	−0.52 ± 7.5 −3(−6–4)	0.02 *

**Table 4 nutrients-15-04057-t004:** Skin and gastrointestinal symptoms in intention-to-treat population at week 4. Each value expressed as number (%) of participants concerned or probability. CMF: cow milk formula; GMF: goat milk formula.

Skin and Gastrointestinal Symptoms	Total (*n* = 64)	GMF (*n* = 33)	CMF (*n* = 31)	*p*
Skin lesions				
Yes	4 (100.0)	2 (50.0)	2 (50.0)	0.85
Atopic dermatitis				
Yes	1 (25.0)	1 (25.0)	0 (0.0)	0.24
Rash				
Yes	0 (0.0)	0 (0.0)	0 (0.0)	
Dry skin				
Yes	3 (75.0)	1 (25.0)	2 (50.0)	0.70
Gastrointestinal symptoms				
Constipation				
Yes	16 (27.6)	12 (38.7)	4 (14.8)	0.042
No	42 (72.4)	19 (61.2)	23 (85.1)	
Missing data	6			
Regurgitation				
Yes	25 (43.1)	11 (35.4)	14 (51.8)	0.20
No	33 (56.9)	20 (64.5)	13 (48.1)	
Missing data	6			
Frequency of regurgitation				
Never	27 (50.9)	15 (55.5)	12 (46.1)	0.60
<50% of feedings	17 (32.2)	7 (25.9)	10 (38.5)	
≥50% of feedings	3 (5.6)	1 (3.7)	2 (7.7)	
Every feeding	6 (11.3)	4 (14.9)	2 (7.7)	
Missing data	11			
Crying during feeding				
Never	44 (80.0)	24 (82.7)	20 (76.9)	0.29
<50% of feedings	8 (14.5)	3 (10.4)	5 (19.2)	
≥50% of feedings	1 (1.8)	0 (0.0)	1 (3.9)	
Every feeding	2 (3.7)	2 (6.9)	0 (0.0)	
Missing data	9			
Abdominal distension				
Yes	11 (18.9)	8 (25.8)	3 (11.1)	0.15
No	47 (81.0)	23 (74.2)	24 (88.8)	
Missing data	6			
Colic pain				
Yes	11 (18.9)	14 (42.4)	11 (35.4)	0.56
No	47 (60.9)	19 (57.5)	20 (64.5)	
Missing data	6			

## Data Availability

Data sets are available upon reasonable request.

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
