# Peer review of "Whole Goat Milk-Based Formula versus Whey-Based Cow Milk Formula: What Formula Do Infants Enjoy More?—A Feasibility, Double-Blind, Randomized Controlled Trial"

_nutrients, 2023, doi:10.3390/nu15184057_

Round 1

Reviewer 1 Report

Comments for authors

This is an interesting, slightly unusual manuscript from a mult center study in France.  Below are a number of questions and comments which I would like to see addressed.

1.      Introduction, page 2 lines 84-86.  The authors report the important study of He et al, and state that there were “similar levels of growth”.  I think as growth outcomes are so important in determining suitability of infant formula we should be given more information here relating to actual growth values and p values that hopefully show no significant differences.

2.      Introduction, page 2 lines 92-24. Similar to the above comments I think it would be good to have a few more details and numbers relating to the FGIDs in this study(Grant).

3.      Methods, Page 3, lines 142-149. The authors quite rightly state their primary outcome measure, being the BEBQ which they say is derived from the CEBQ validated for older ages.  Whilst I cannot easily see how it could be done, but are there any attempted validation studies in infants? If so, please describe, if not please state so.

4.      Methods, page 4 lines 168-174.  While it is good to see some details around the sample size calculation, I would like to ger some clarification here. The authors state that they are aiming to detect a difference of 0.27 between groups. Where did that number come from? Also I am confused about the statement that “because this was a feasibility study…..lowered to 50”.  I don’t understand this.  The fact that this is a feasibility study does not, in my view, justify reducing the numbers to below those required to detect a difference.  If the intention was to state that for practical/funding/ time reasons the numbers were reduced this should be stated. Also, and importantly, the difference that could be detected with n=50 should be stated.

5.      Results, page 4 lines 190-192. I am very aware of how COVID-19 impacted on studies like these and the authors should be congratulated for persevering with their work and completing the study. No changes requested, just a comment.!

6.      Results, page 5, table 1. It may be a function of how I am viewing the draft manuscript but the responses “alcohol consumption during pregnancy “ seem not aligned correctly. Please just check.

7.      Results, page 7, table 3. Again it may be a function of the way I am viewing the draft manuscript but the data in this table seem to be badly aligned. Please just check.

8.      Discussion, page 9 lines 313-315.  The authors comments about “lacked the information to precisely calculate the number of subjects required” reinforces my comment above in point 4.

9.      Statements, page 11, lines 348-365. I note that two of the authors are/were employed by the funder. However, this is clearly stated and declared and should be an issue.

Author Response

Thank you for your reviewing.

  1. Introduction, page 2 lines 84-86.  The authors report the important study of He et al, and state that there were “similar levels of growth”.  I think as growth outcomes are so important in determining suitability of infant formula we should be given more information here relating to actual growth values and p values that hopefully show no significant differences.

Thank you for this remark. More details have been provided (line 86-92)

  1. Introduction, page 2 lines 92-24. Similar to the above comments I think it would be good to have a few more details and numbers relating to the FGIDs in this study(Grant).

More details are now given in the introduction section.

  1. Methods, Page 3, lines 142-149. The authors quite rightly state their primary outcome measure, being the BEBQ which they say is derived from the CEBQ validated for older ages.  Whilst I cannot easily see how it could be done, but are there any attempted validation studies in infants? If so, please describe, if not please state so.

Thank you for this question. The BEBQ has been validated in infants (Llewellyn et al. / Appetite 57 (2011) 388–396). This point has been added in the method section.

  1. Methods, page 4 lines 168-174.  While it is good to see some details around the sample size calculation, I would like to ger some clarification here. The authors state that they are aiming to detect a difference of 0.27 between groups. Where did that number come from? Also I am confused about the statement that “because this was a feasibility study…..lowered to 50”.  I don’t understand this.  The fact that this is a feasibility study does not, in my view, justify reducing the numbers to below those required to detect a difference.  If the intention was to state that for practical/funding/ time reasons the numbers were reduced this should be stated. Also, and importantly, the difference that could be detected with n=50 should be stated.

Additional details have been provided, and a power calculation has been performed: reducing the number of patients to 50 per group resulted in a 60% decrease in power.

  1. Results, page 4 lines 190-192. I am very aware of how COVID-19 impacted on studies like these and the authors should be congratulated for persevering with their work and completing the study. No changes requested, just a comment.!

Thank you for this comment

  1. Results, page 5, table 1. It may be a function of how I am viewing the draft manuscript but the responses “alcohol consumption during pregnancy “ seem not aligned correctly. Please just check.

Thank you. This has been modified

  1. Results, page 7, table 3. Again it may be a function of the way I am viewing the draft manuscript but the data in this table seem to be badly aligned. Please just check.

Thank you for this remark. I hope it’s better now.

  1. Discussion, page 9 lines 313-315.  The authors comments about “lacked the information to precisely calculate the number of subjects required” reinforces my comment above in point 4.

Yes, thank you for this comment

  1. Statements, page 11, lines 348-365. I note that two of the authors are/were employed by the funder. However, this is clearly stated and declared and should be an issue.

Some precision has been added in the conflict of interest section to precise the implication of these 2 co-authors: “CP was involved in writing the protocol. SG was involved in the data interpretation and in writing the manuscript. Neither CP nor SG were involved in recruitment, data collection, or statistical analysis.”

Reviewer 2 Report

Dear editors and authors, thank you very much for allowing me to review this manuscript entitled  “Whole goat milk-based formula versus whey-based cow milk formula: what formula do infants enjoy more? – A feasibility, double-blind, randomized controlled trial”

I will make mis comments below:

1.- Objective and design:

The objective and the initial design seem correct, although only 28 days of follow-up seems a short time to reach the conclusions provided.

2.- Sample size and characteristics of the subjects:

2.1.- The size of the sample is clearly small and insufficient to be able to affirm the conclusions. The authors try to justify the sample size due to the COVID pandemic, but although we can accept it, the sample size is such an important piece of information that the authors should be very conservative in their conclusions.

2.2.- The percentage of children with mixed lactation or the amount of breast milk they drink at the beginning or at the end is not stated (at first they say that less than 50%, they refer, but this data is not very informative). Nor do they talk about the schedule of intakes or the frequency of the same, very important data when evaluating the proposed tests.

3.- Results:

3.1.- Regarding the statistical aspects, in the comparison of test scores, it would be better to compare them using medians (and not averages). The Wilcoxon test uses order and not quantity.

3.2.- The differences found are so small that the "real" effects on children could be discussed.

3.3.- Another point to take into account is the scant follow-up of the children. They are only followed for a month and this must be taken into account when drawing conclusions.

3.4.- An important aspect is that the gastrointestinal alterations of the children were not measured at the beginning, as indicated by the authors. In addition, they say that the differences in constipation have to be taken with caution because they are the opinions of the parents, but aren't the tests analyzed also? (line 309).

4.- Discussion and conclusions:

4.1.- The difference in milk fat can justify some aspects of those found, but it would be convenient to discuss whether this increase in fat is good, or not, for children.

4.2. If QUALIN is not validated (Line 321), the conclusions on quality of life should not be so strong.

4.3.- In the bibliography: only 11 of 33 citations are from the last 5 years (it would be interesting to cite: Jankiewicz M, van Lee L, Biesheuvel M, Brouwer-Brolsma EM, van der Zee L, Szajewska H. The Effect of Goat -Milk-Based Infant Formulas on Growth and Safety Parameters: A Systematic Review and Meta-Analysis. Nutrients. 2023 Apr 27;15(9):2110. doi: 10.3390/nu15092110).

4.4.- Perhaps it would be important to make clear the relationship of two authors with the company that promotes this research.

5.- Minor aspects:

- The first time the term “Intention-to-treat (ITT)” appears, it is on line 176, and not on 195.

- The first time the term “The perprotocol (PP) population” appears is on line 182, and not on line 197.

Only minor changes to the English of the text would be necessary.

Author Response

1.- Objective and design:

The objective and the initial design seem correct, although only 28 days of follow-up seems a short time to reach the conclusions provided.

The conclusions of the manuscript and the abstract have been tempered.

2.- Sample size and characteristics of the subjects:

 2.1.- The size of the sample is clearly small and insufficient to be able to affirm the conclusions. The authors try to justify the sample size due to the COVID pandemic, but although we can accept it, the sample size is such an important piece of information that the authors should be very conservative in their conclusions.

The conclusion has been tempered.

2.2.- The percentage of children with mixed lactation or the amount of breast milk they drink at the beginning or at the end is not stated (at first they say that less than 50%, they refer, but this data is not very informative). Nor do they talk about the schedule of intakes or the frequency of the same, very important data when evaluating the proposed tests.

Thank you for this observation. Infants were predominantly required to be formula-fed to be included. Indeed, the exact percentage distribution between formula feeding and breastfeeding was not collected for this study conducted in outpatient pediatrics. Therefore, we chose to limit the number of variables to be collected by the physician. Additionally, the breastfeeding rate in France at 4 months is low (<30%) (chrome-extension://efaidnbmnnnibpcajpcglclefindmkaj/https://drees.solidarites-sante.gouv.fr/sites/default/files/2020-08/er958.pdf)

3.- Results:

3.1.- Regarding the statistical aspects, in the comparison of test scores, it would be better to compare them using medians (and not averages). The Wilcoxon test uses order and not quantity.

Clarifications have been provided on this point in the discussion (lines 314-319). Median and iqr have been added in the tables.

3.2.- The differences found are so small that the "real" effects on children could be discussed.

The conclusion has been modified accordingly (lines 362-364)

3.3.- Another point to take into account is the scant follow-up of the children. They are only followed for a month and this must be taken into account when drawing conclusions.

This point has also been added (line 357)

3.4.- An important aspect is that the gastrointestinal alterations of the children were not measured at the beginning, as indicated by the authors. In addition, they say that the differences in constipation have to be taken with caution because they are the opinions of the parents, but aren't the tests analyzed also? (line 309).

Constipation was more frequently reported in the GMF group compared to the CMF group (table 4).

4.- Discussion and conclusions:

4.1.- The difference in milk fat can justify some aspects of those found, but it would be convenient to discuss whether this increase in fat is good, or not, for children.

This point has been discussed in the discussion section: “Beyond a potential impact on sensorial characteristics of the formulas, the increased proportion of milk in the whole goat milk formula supplies higher levels of sn-2 palmitic acid, Milk Fat Globule Membrane, cholesterol and short and medium-chain fatty acids, which may influence the digestion of lipids”

4.2. If QUALIN is not validated (Line 321), the conclusions on quality of life should not be so strong.

Modifications have been done in the discussion section, line 321

4.3.- In the bibliography: only 11 of 33 citations are from the last 5 years (it would be interesting to cite: Jankiewicz M, van Lee L, Biesheuvel M, Brouwer-Brolsma EM, van der Zee L, Szajewska H. The Effect of Goat -Milk-Based Infant Formulas on Growth and Safety Parameters: A Systematic Review and Meta-Analysis. Nutrients. 2023 Apr 27;15(9):2110. doi: 10.3390/nu15092110).

Thank you for this point. This article has been added in the introduction section

4.4.- Perhaps it would be important to make clear the relationship of two authors with the company that promotes this research.

Thank you for this observation. Infants were predominantly required to be formula-fed to be included. Indeed, the exact percentage distribution between formula feeding and breastfeeding was not collected for this study conducted in outpatient pediatrics. Therefore, we chose to limit the number of variables to be collected by the physician. Additionally, the breastfeeding rate in France at 4 months is low (<30%) (chrome-extension://efaidnbmnnnibpcajpcglclefindmkaj/https://drees.solidarites-sante.gouv.fr/sites/default/files/2020-08/er958.pdf)

5.- Minor aspects:

- The first time the term “Intention-to-treat (ITT)” appears, it is on line 176, and not on 195.

Thank you. The modification has been made

- The first time the term “The perprotocol (PP) population” appears is on line 182, and not on line 197.

Thank you. The modification has been made
